# Current knowledge of vector-borne zoonotic pathogens in Zambia: A clarion call to scaling-up "One Health" research in the wake of emerging and re-emerging infectious diseases

Benjamin Mubemba[1,2], Monicah M. Mburu[3], Katendi Changula[4], Walter Muleya[5], Lavel C. Moonga[6], Herman M. Chambaro[7], Masahiro Kajihara[8], Yongjin Qiu[9], Yasuko Orba[7,10], Kyoko Hayashida[6,10], Catherine G. Sutcliffe[11,12], Douglas E. Norris[13], Philip E. Thuma[3], Phillimon Ndubani[3], Simbarashe Chitanga[14,15,16], Hirofumi Sawa[7,9,10,17,18,19,20], Ayato Takada[8,17,18]*, Edgar Simulundu[3,17]*

1 Department of Wildlife Sciences, School of Natural Resources, Copperbelt University, Kitwe, Zambia, 2 Department of Biomedical Sciences, School of Medicine, Copperbelt University, Ndola, Zambia, 3 Macha Research Trust, Choma, Zambia, 4 Department of Paraclinical Studies, School of Veterinary Medicine, University of Zambia, Lusaka, Zambia, 5 Department of Biomedical Sciences, School of Veterinary Medicine, University of Zambia, Lusaka, Zambia, 6 Division of Collaboration and Education, International Institute for Zoonosis Control, Hokkaido University, Sapporo, Japan, 7 Division of Molecular Pathobiology, International Institute for Zoonosis Control, Hokkaido University, Sapporo, Japan, 8 Division of Global Epidemiology, International Institute for Zoonosis Control, Hokkaido University, Sapporo, Japan, 9 Division of International Research Promotion, Hokkaido University International Institute for Zoonosis Control, Sapporo, Japan, 10 International Collaboration Unit, International Institute for Zoonosis Control, Hokkaido University, Sapporo, Japan, 11 Department of International Health, Johns Hopkins Bloomberg School of Public Health, Baltimore, Maryland, United States of America, 12 Department of Epidemiology, Johns Hopkins Bloomberg School of Public Health, Baltimore, Maryland, United States of America, 13 The W. Harry Feinstone Department of Molecular Microbiology and Immunology, The Johns Hopkins Malaria Research Institute, Johns Hopkins Bloomberg School of Public Health, Baltimore, Maryland, United States of America, 14 Department of Paraclinical Studies, School of Veterinary Medicine, University of Namibia, Windhoek, Namibia, 15 Department of Biomedical Sciences, School of Health Sciences, University of Zambia, Lusaka, Zambia, 16 School of Life Sciences, College of Agriculture, Engineering and Sciences, University of KwaZulu-Natal, Durban, South Africa, 17 Department of Disease Control, School of Veterinary Medicine, University of Zambia, Lusaka, Zambia, 18 Africa Centre of Excellence for Infectious Diseases of Humans and Animals, University of Zambia, Lusaka, Zambia, 19 Global Virus Network, Baltimore, Maryland, United States of America, 20 One Health Research Center, Hokkaido University, Sapporo, Japan

* atakada@czc.hokudai.ac.jp (AT); esikabala@yahoo.com (ES)

**Data Availability Statement:** All relevant data is available within the manuscript.

## Abstract

### Background

Although vector-borne zoonotic diseases are a major public health threat globally, they are usually neglected, especially among resource-constrained countries, including those in sub-Saharan Africa. This scoping review examined the current knowledge and identified research gaps of vector-borne zoonotic pathogens in Zambia.

### Methods and findings

Major scientific databases (Web of Science, PubMed, Scopus, Google Scholar, CABI, Scientific Information Database (SID)) were searched for articles describing vector-borne (mosquitoes, ticks, fleas and tsetse flies) zoonotic pathogens in Zambia. Several mosquito-

**Funding:** HS is supported by the Japan Program for Infectious Diseases Research and Infrastructure (JP21wm0125008 and JP21wm0225017), and the Japan Agency for Medical Research and Development. AT is supported by the Japan International Cooperation Agency (JICA) within the framework of the Science and Technology Research Partnership for Sustainable Development (SATREPS) (JP20jm0110019). The funders had no role in study design, data collection and analysis, decision to publish, or preparation of the manuscript.

**Competing interests:** The authors have declared that no competing interests exist.

borne arboviruses have been reported including Yellow fever, Ntaya, Mayaro, Dengue, Zika, West Nile, Chikungunya, Sindbis, and Rift Valley fever viruses. Flea-borne zoonotic pathogens reported include *Yersinia pestis* and *Rickettsia felis*. *Trypanosoma* sp. was the only tsetse fly-borne pathogen identified. Further, tick-borne zoonotic pathogens reported included Crimean-Congo Haemorrhagic fever virus, *Rickettsia* sp., *Anaplasma* sp., *Ehrlichia* sp., *Borrelia* sp., and *Coxiella burnetii*.

## Conclusions

This study revealed the presence of many vector-borne zoonotic pathogens circulating in vectors and animals in Zambia. Though reports of human clinical cases were limited, several serological studies provided considerable evidence of zoonotic transmission of vector-borne pathogens in humans. However, the disease burden in humans attributable to vector-borne zoonotic infections could not be ascertained from the available reports and this precludes the formulation of national policies that could help in the control and mitigation of the impact of these diseases in Zambia. Therefore, there is an urgent need to scale-up "One Health" research in emerging and re-emerging infectious diseases to enable the country to prepare for future epidemics, including pandemics.

## Author summary

Despite vector-borne zoonoses being a major public health threat globally, they are often overlooked, particularly among resource-constrained countries in sub-Saharan Africa, including Zambia. Therefore, we reviewed the current knowledge and identified research gaps of vector-borne zoonotic pathogens in Zambia. We focussed on mosquito-, tick-, flea- and tsetse fly-borne zoonotic pathogens reported in the country. Although we found evidence of circulation of several vector-borne zoonotic pathogens among vectors, animals and humans, clinical cases in humans were rarely reported. This suggests sparse capacity for diagnosis of vector-borne pathogens in healthcare facilities in the country and possibly limited awareness and knowledge of the local epidemiology of these infectious agents. Establishment of facility-based surveillance of vector-borne zoonoses in health facilities could provide valuable insights on morbidity, disease severity, and mortalities associated with infections as well as immune responses. In addition, there is also need for increased genomic surveillance of vector-borne pathogens in vectors and animals and humans for a better understanding of the molecular epidemiology of these diseases in Zambia. Furthermore, vector ecology studies aimed at understanding the drivers of vector abundance, pathogen host range (i.e., including the range of vectors and reservoirs), parasite-host interactions and factors influencing frequency of human-vector contacts should be prioritized. The study revealed the need for Zambia to scale-up One Health research in emerging and re-emerging infectious diseases to enable the country to be better prepared for future epidemics, including pandemics.

## Introduction

Zoonoses are infectious diseases that are spread from animals to humans or vice versa. It is estimated that more than 60% of emerging human infectious diseases have their origins in

animal populations [1], with climatic shifts and anthropogenic land use changes implicated as major drivers of this emergence. These drivers lead to the reduction of the biogeographical distances between the vector and/or zoonotic pathogens or the infected animal populations and humans, allowing for pathogen transmission and subsequent human zoonotic infections [2,3]. In sub-Saharan Africa, zoonoses that have emerged in the last four decades are mostly viral and bacterial agents that have caused disease burdens of varying proportions in humans [2]. The impact of zoonoses on public health systems globally can be far reaching. More generally, it is estimated that more than 2 million human deaths and 2.4 billion cases of illness every year globally are attributed to zoonoses [4]; a situation requiring urgent intervention from the public-health and scientific communities.

Continued surveillance of zoonotic pathogens is crucial in building public health systems that translate into effective strategies for prevention and control of zoonotic diseases [5]. Ideally, this will include robust research infrastructure both in expertise and laboratory capacity as well as implementation of a "One Health" approach which should be multi-sectoral and multidisciplinary in nature as well as operationally [6]. Implementing such an approach will not only improve the response of public health systems in mitigating the impact of zoonoses, but will also benefit the animal, environmental and wildlife sectors which are critically important components of the economy [5]. The current challenges in the fight against zoonoses in resource-constrained countries including Zambia are to a large extent due to lack of appropriate diagnostics and limited knowledge about zoonotic pathogens, resulting in the misdiagnosis of conditions unfamiliar to medical practitioners. This is a reflection of insufficient knowledge of what pathogens are in circulation, compounded by an absence of appropriate diagnostics. It has been shown that across Africa, there tends to be under/misdiagnosis of less known infections accompanied by overdiagnosis of the most common ones [7].

In Zambia, several zoonotic pathogens have been reported in different ecological landscapes, but their etiological impact on human diseases remains unclear. Currently, information on the disease burden attributable to vector-borne and emerging or re-emerging zoonoses in humans is obscure. For example, despite a number of zoonotic pathogens (bacterial and arboviral zoonoses) being recognized as etiologic agents of febrile illnesses [8], they are not routinely screened for in the differential diagnosis of febrile illnesses in most healthcare facilities, largely due to lack of diagnostic capacity as well as a poor understanding of the local epidemiology of these pathogens among health practitioners.

Here, we reviewed studies reporting zoonoses or their etiological agents in Zambia focussing on mosquito-, tick-, flea- and tsetse-borne pathogens, and identified knowledge gaps that should be addressed in future studies. This study further aimed at stimulating interest in scaling-up research in emerging and re-emerging pathogens in order to create awareness among public health practitioners and drive policy change towards response, diagnosis and management of zoonotic diseases in the country. Currently, genomic surveillance of these emerging and re-emerging zoonotic pathogens in humans is hugely lacking as earlier studies were mostly serological. Thus, there is an urgent need to provide complementary genomic data necessary to inform and guide the response, diagnosis and management of zoonotic diseases in the country.

## Methods

This review was based on publications indexed in the Web of Science, PubMed, Scopus, Google Scholar, CABI or Scientific Information Database (SID), prior to June, 2021. We also included a study "in press" conducted by the authors. Our online search included a combination of keywords such as–"mosquito-borne", "tick-borne", "flea-borne", "tsetse-borne" and/or

"zoonoses in Zambia", "Vector-borne disease" and "arboviral diseases in Zambia" and filtering the literature to include only publications describing their occurrence in Zambia. We further searched these electronic databases for literature reporting the occurrence of these diseases in animals and humans in Zambia, Southern Africa and in some instances at a continental level to build context for the regional perspective. Finally, we expanded our search to review references cited in the searched publications to increase coverage of the literature that was relevant to the preparation of this review. Through this filtering process, a total of 39 articles reporting detection of vector-borne zoonoses in Zambia were identified for the preparation of the review. Four main categories for this review were prepared to highlight mosquito-borne, tick-borne, flea-borne and tsetse-borne zoonotic pathogens reported in Zambia and further discussed to identify research gaps to inform future investigations.

## Results

### Mosquito-borne Zoonotic viruses

Mosquito-borne viruses of medical and veterinary importance that have been reported in Zambia mainly belong to four virus genera; *Flavivirus*, *Alphavirus*, *Orthobunyavirus* and *Phlebovirus*.

**Flaviviruses.** The genus *Flavivirus* consists of positive single-stranded RNA enveloped viruses. Notable members of public health significance include; Yellow fever virus (YFV), Ntaya virus (NTAV), Dengue virus (DENV), Zika virus (ZIKV), West Nile virus (WNV), Japanese encephalitis and several other viruses which mostly cause febrile and neurological manifestations in humans [9].

Yellow fever virus (YFV), the prototype virus in the genus *Flavivirus*, is a re-emerging threat with more than 508 million people at risk of infection in tropical African countries where the virus remains endemic [10]. The virus is mainly transmitted by *Haemagogus* and *Aedes* mosquitoes either through the sylvatic or urban cycles of transmission [11]. Clinically, YFV infection is characterized by fever, headache, jaundice, muscle pain, nausea, vomiting and fatigue [11]. The earliest mention of YFV in Zambia dates back to the early 1940s from seroprevalence studies that revealed low exposure of people in present-day North-Western province [12]. More recently, a study in Western and North-Western provinces conducted in 2015 revealed a seroprevalence of 0.2% (3,325 study participants), suggesting a considerably low active circulation of the virus [13].

Ntaya virus (NTAV) was first isolated in western Uganda in the early 1950s [14]. Though a limited number of NTAV infections have been recorded across the globe, notable consistent clinical symptoms observed in humans include fever, headache and neurological symptoms [15]. Though not confirmed, *Culex* mosquitoes are suspected to be vectors of NTAV [16]. Serological evidence of NTAV in Zambia was documented in 1977 among international travellers that visited Zambia [15]. This study is the only report of the pathogen in the country. Elsewhere, serological evidence of infection has been reported in migratory birds as well as farmed animals, suggesting a potential reservoir role of these hosts [17,18].

Dengue virus has in the recent past been viewed as one of the most serious re-emerging vector-borne zoonosis with more than 390 million infections per year in the Americas, tropical Africa and Asia [19]. The virus is transmitted by mosquitoes of the genus *Aedes* [20]. Infections in humans can either be asymptomatic or symptomatic. In apparent and mild clinical infections, they are characterised by headache, rash, myalgia, arthralgia, and fever. Hepatitis, neurological disorders, myocarditis, and shock have been observed in severe infections [21]. Serological evidence of DENV in Zambia was first reported among European expatriates and travellers that had visited Zambia between 1987 and 1993 [22]. Further reports of DENV were

in North-Western and Western provinces of Zambia with a prevalence rate of 4.1% (149/3624) [23], and more recently in Central Province [24]. However, there are currently no reports of human clinical cases.

Sylvatic and urban cycling exist for ZIKV transmission, although in Africa, sylvatic cycle seems to be the main transmission route with mosquitoes of the genus *Aedes* being the principal vectors [25]. Clinically, human infections are characterised by mild symptoms of fever, rash, joint and muscle pain [26]. In pregnant women, the infection can lead to microcephaly in the infants [26]. To date, there are no documented human clinical cases of ZIKV infection in Zambia. However, its possible presence in Zambia has been demonstrated through serosurveys with reported prevalence rate of 6% (217/3625) in humans in Western and North-Western provinces [27], and 10.8% (23/214) in Lukanga swamps of Central Province [24]. In addition, neutralizing antibodies against ZIKV were detected in 34.4% (33/96) of African green monkeys and baboons inhabiting different ecosystems in Zambia, a finding that was indicative of a recent active infection and possible sylvatic maintenance of the pathogen in the country [28].

Generally, WNV infections in humans are predominantly asymptomatic. However, fever, neurological disease and death have been reported in severe symptomatic cases [29]. The first report of WNV in Zambia was in 2015 when a seroprevalence study conducted showed a prevalence rate of 10.3% (370/3625) for WNV among humans in North-Western and Western provinces [30]. Genetic evidence of the circulation of the virus has also been shown in *Culex quinquefasciatus* mosquitoes collected in Western Province [31] as well as in crocodiles [32] with lineages 2 and 1a reported in the respective studies. The lineage 2 strain reported in Zambia was shown to be closely related to strains circulating in South Africa, which have been reported to be associated with neurological disease in humans [31].

**Alphaviruses.**   Alphaviruses are enveloped, positive-sense RNA viruses with all human infective viruses being mosquito-borne. A notable member in this group is chikungunya virus (CHIKV) that causes chikungunya fever. It is an emerging mosquito-borne disease that is principally transmitted by *Aedes* mosquitoes [33]. The disease shares many clinical similarities with DENV including acute onset of fever, joint pains, headache and fatigue. Previously, chikungunya fever was considered a self-limiting disease. However, severe complications, including acute viral hepatitis and death, have been reported in elderly patients and people with chronic comorbidities [34]. The first suspected reports of chikungunya fever in Zambia were recorded in the colonial era on the Copperbelt Province along the Kafue River basin with most cases having classical symptoms of CHIKV infection coupled with generalised superficial lymphadenopathy [35]. However, in 1999, antibodies against Sindbis virus (SINV) were detected in 0.1% (1/670) of German travellers that had spent at least 4 months in Zambia [36], indicating the possible presence of this pathogen in the country. Recently, a study conducted in 2016 among residents of Lukanga swamps in Central Province, revealed a 36.9% (79/214) sero-positivity for CHIKV and 19.6% (42/214) for Mayaro virus (MAYV) [24]. Interestingly and in contrast, a recent study that screened for alphaviruses using 9,699 mosquitoes collected from other geographical areas in Zambia (2014–2017) could not detect known alphaviruses (CHIKV, O'nyong-nyong virus (ONNV), SINV etc.) [37]. One possible explanation for this discrepancy were sampling sites for the vector-based study that did not include Lukanga swamps. It is also possible that the species composition of the tested mosquitoes may have contributed to the discrepancy. For instance, the majority of the captured mosquitoes were *Culex* and alphaviruses such as CHIKV and O'nyong-nyong virus (ONNV) could be missed as their principle vectors are Aedes and Anopheles mosquitoes, respectively. Nevertheless, it is imperative that future genomic surveillance studies should also target to screen alphavirus vectors in Lukanga swamps where human exposure to these pathogens has been documented.

**Phleboviruses.**   Rift Valley fever virus (RVFV) infections in humans are characterised by a wide range of symptoms, which may include fever, myalgia, arthralgia, headache encephalitis, haemorrhage, hepatitis, and ocular pathology/retinitis [38]. Human zoonotic infections are usually preceded by the disease in wild and domestic animals [38]. In Zambia RVFV was first reported in humans in Chisamba District, Central Province, in 1984 where it caused some deaths [39]. Further evidence of the disease was provided in subsequent studies conducted in the early 1990s that indicated that cattle-dominated regions with higher amounts of rainfall had high seropositivity in the herds tested [39,40]. However, for the past three decades, no reports of outbreaks have been noted in humans or animals defying the epidemic resurgence pattern of 10–15 years as previously proposed [41]. However, a recent study suggests silent circulation of RVFV in cattle herds [42] corroborating with evidence suggesting that the lack of natural physical barriers between Zambia and other countries in the region that have reported outbreaks in the recent past, cross-border livestock trade and the presence of competent vectors present a real threat of RVFV in Zambia [43–45].

## Tick-borne zoonotic pathogens

**Viruses.**   Notable tick-borne viral zoonoses include tick-borne encephalitis virus (TBEV) and Crimean–Congo haemorrhagic fever virus (CCHFV), with CCHFV being the most recognised tick-borne viral zoonosis globally. Infections due to CCHFV in humans are characterised by headache, fever, joint pain, stomach pain, and vomiting [46]. Currently, there is only a single report of CCHFV in Zambia, with IgG antibodies being detected in 8.4% (88/1,047) of cattle [47]. The pathogen genome was detected in 3.8% (11/290) of *Hyalomma* ticks, the principal vector of CCHFV [47]. Genomic analyses revealed that one of the detected viruses was a genetic reassortant between African and Asian strains, intimating possible dissemination of CCHFV across continents. There is currently no report of tick-borne viral infections in humans in Zambia.

**Bacteria.**   A number of tick-borne bacterial fever-inducing pathogens of zoonotic significance have been described to circulate in different hosts and vectors in Zambia. These belong to bacterial families of *Rickettsiaceae*, *Anaplasmataceae*, *Spirochaetaceae* and *Coxiellaceae*. Clinically, symptoms associated with tick-borne bacterial zoonoses are usually non-specific, making their diagnosis and treatment challenging and can include; fever, headache, body rash, nausea, joint and muscle pains [48].

In a study targeted to screen zoonotic pathogens in free-ranging nonhuman primates (NHPs) inhabiting various ecological landscapes in Zambia, zoonotic *Anaplasma phagocytophilum* was molecularly detected in 13.6% (12/88) of samples screened [49]. In another study, Vlahakis *et al.*, (2018) [50], reported the first molecular identification and characterization of canine-associated zoonotic *Anaplasma* species in Zambia. The prevalence rate was 9% (27/301) for the dog population screened [50]. In addition, *Ehrlichia canis* and *Anaplasma platys* were genetically detected in dogs screened in Lusaka province [51].

*Rickettsia* species that have been identified in ticks and NHPs thus far in Zambia include *R. africae*, *R. raoultii*, *R. conorii*, *R. massiliae*, *R. lusitaniae*, *R. hoogstraalii*, *R. aeschlimannii*-like, and some unidentified *Rickettsia* species [52–56]. Notably, most of the *Rickettsia* species detected in the country are known human pathogens. Though no human clinical cases have been reported so far, serological evidence for infection with *R. conorii* and *R. typhi* was reported in 1999 with prevalence rates of 16.7% (63/377) and 5.0% (19/377), respectively [57].

Human borreliosis is characterised by erythema migrans, fever, chills, headache, fatigue, muscle and joint aches, and swollen lymph nodes. Only one case of human relapsing fever caused by *Borrelia* spp. has been documented in Zambia involving a patient with a history of

tick bite from a soft tick (*Ornithodoros faini*) known to infest cave dwelling fruit bats (*Rousettus aegyptiacus*) [58]. The isolated *Borrelia* spp. was more closely related to New World relapsing fever *Borreliae*, indicating the possibility of these zoonotic infections being imported from afar places, possibly through migratory bats. Despite being a single case of human borreliosis, these results suggest that regions in Zambia with overlapping ecological landscapes of *O. faini* ticks and *R. aegyptiacus* bats may be at risk of transmission of tick-borne relapsing fever to humans [58].

*Coxiella burnetii*, the causative agent of Q fever in humans is a zoonotic agent with ubiquitous global distribution. Symptoms in human infections include; high fever, chills or sweats, a cough, chest pain while breathing, headache, diarrhoea, nausea, abdominal discomfort and in some instances jaundice. It has been detected in a number of tick species known to infest livestock such as *Rhipicephalus*, *Amblyomma* and *Haemaphysalis* [59]. It was first recognised in Zambia among livestock (cattle and goats) in Chama, Chongwe and Petauke districts through molecular typing [60]. More recently, Chitanga *et al.*, (2018) provided the first genetic evidence of *C. burnetii* circulating in Zambian dogs and rodents [61]. In humans, about two decades ago, serological evidence of Q fever infection was demonstrated in Zambia with a prevalence rate of 8.2% [57,62]. These studies show that the risk of transmission of *C. burnetii* to humans especially among livestock farming communities and pet owners is conspicuous and presents a real but unappreciated threat [63].

## Flea-borne zoonotic pathogens

Flea-borne zoonoses are emerging or re-emerging worldwide, and their incidence is on the rise. Furthermore, their distribution and that of their vectors is shifting and expanding [64]. However, their public health significance in Zambia has not been sufficiently elucidated partly due to lack of epidemiological surveillance despite the abundance of cosmopolitan flea vectors. To date only *Yersinia pestis* and *Rickettsia felis* have been reported in Zambia [65–67].

Recognized for its previous global pandemics called "Black death", *Y. pestis* was first documented to cause human infections in Zambia in 1914 with a number of subsequent outbreaks recorded [68]. Bubonic plague is the most common disease form of the plague in Zambia, and is a rodent-flea-borne re-emerging zoonosis characterised by acute onset of fever, headache and body malaise. In Zambia, it is endemic in Namwala (Southern Province), Sinda and Nyimba districts (Eastern Province). A study conducted between 2015 and 2016 in Sinda and Nyimba districts showed the circulation of *Y. pestis* among multiple hosts including humans, fleas, shrews and rodents [69]. Sequences of strains isolated from these hosts were similar, indicating a multi-host zoonotic circulation and they demonstrated very close evolutionary relationships with Antiqua strains from the Democratic Republic of Congo and Kenya. In addition, serological studies conducted in the same area showed that 16.0% (4/25) of rodents, 16.7% (1/6) of shrews and 6.0% (5/83) of goats were positive for IgG antibodies against Fraction 1 antigen of *Y. pestis*, a result supporting the idea of active circulation of the bacterium in the geographic area [70]. Risk factors associated with human outbreaks in Zambia included heavy rains leading to a huge surge of rodents and fleas, hunting, unhygienic human habitats, preparation and consumption of rodents [71,72].

Flea-borne spotted fever caused by *Rickettsia felis* is one of the neglected emerging rickettsioses. Cat fleas have been considered the hosts and biological vectors of *R. felis* with dogs and cats acting as mammalian reservoirs [73,74]. *R. felis* is transovarially and transtadially maintained in cat fleas [75]. *R. felis* is classified in the spotted fever group (SFG) rickettsiosis based on clinical disease manifestation. The first human case of *R. felis* infection was reported in 1994 in Texas, USA [76]. Since then, it remained neglected until it emerged as a cause of febrile

illness in sub-Saharan Africa [77]. In Zambia, the bacterium was recently detected in 4.7% (7/150) of dogs and 11.3% (12/106) of *Mastomys* species of rodents [67]. Additionally, 3.7% (2/53) of the sampled cat fleas collected from dogs were positive for *R. felis* [67]. Another potential cause of flea-borne rickettsiosis in humans is *R. asembonensis,* which is similarly transmitted from fleas to humans. Although its pathogenicity in humans is unknown, several reports indicated the infection is associated with acute febrile illness [78]. The prevalence of *R. asembonensis* in cat fleas collected from Zambian dogs was reported to be high showing 41.5% (22/53) [67]. Importantly, the DNA of *R. asembonensis* was also detected in two human blood samples from Zambia out of 1,153 screened. This suggests possible human infection by *R. asembonensis* possibly through cat flea bite [79].

## Tsetse-borne pathogens

Trypanosomiasis has been endemic in Zambia for more than a century and is prevalent in wildlife-livestock ecosystems having geographical overlaps with tsetse-belts, where the fly that vectors these pathogens are found [80]. The causative agent, *Trypanosoma* sp., is the only pathogen transmitted by the biting tsetse flies (*Glossina* spp.) that are endemic to the Luangwa and Zambezi valleys and Kafue National Park (KNP) [80]. Of public health concern in Zambia is the human infective trypanosome *T. b. rhodesiense* which is endemic in Eastern and Southern Africa in various animal hosts and tsetse flies [81]. Recently, Squarre *et al.,* (2020), detected several *Trypanosoma* sp. in multiple wildlife species inhabiting KNP including the human infective *T. b. rhodesiense* [82]. The zoonotic trypanosome has also been reported in both indigenous and exotic dog breeds kept as pets in areas bordering endemic zones, a finding showing the increased possibility of transmission of Human African Trypanosomiasis (HAT) from the wildlife-livestock-tsetse interface to surrounding human communities [83–85]. Evidence is increasing to suggest that more cases of HAT than previously thought occur among communities lying along the tsetse belts. For example, in 2012, HAT was diagnosed in four patients from the Luangwa and Zambezi valleys that presented with febrile illness [86]. Furthermore, a severe case of HAT was also recorded in 2016 in KNP from a patient that presented with persistent fever [87]. Whilst these documented cases may seem to be sporadic, previous passive surveys for HAT indicated that HAT was frequently reported among communities or workers living in tsetse fly-infested wildlife protected areas but was underreported [88,89]. Among the reasons for this underreporting included lack of diagnostic capacity in rural health centres, low awareness levels among health care providers, and frequent misdiagnosis with familiar diseases such as malaria [90]. This further affirms that HAT is still a neglected tropical disease in Zambia [91].

## Pathogen discovery

On the pathogen discovery front, some strides have been made in that a number of novel arboviruses have been detected in the country. Among them include the tick-borne phlebovirus, Shibuyunji virus, which was detected from *Rhipicephalus* ticks in Shibuyunji and Namwala district [92,93]. Additionally, Harima and colleagues (2021), characterised the Mpulungu flavivirus, a novel tick-borne flavivirus isolated from a *Rhipicephalus muhsamae* tick with a typical vertebrate genome signature, suggesting its potential to infect vertebrate hosts [94]. However, studies in both humans and animals associated with the ticks are certainly needed to clarify the veterinary and public health importance of these novel tick-borne viruses. Others include Mwinulunga alphavirus isolated from *Culex quinquefasciatus* mosquitoes [37] and the newly identified Barkedji-like virus; a novel flavivirus detected from *Culex* spp. mosquitoes [95]. Current experimental data so far suggests that both Mwinilunga alphavirus and Barkedji-like virus

fail to replicate on vertebrate cell lines, but it would be interesting to know the implications of this experimental data on host range restriction and whether they have any zoonotic potential [37,95].

## Identified research gaps on vector-borne zoonotic pathogens in Zambia

For mosquito-borne pathogens, we observed that most reports (14/19; 73.6%) were in humans, with only five (26.4%) other studies reporting the pathogens in vectors or other mammalian hosts (Table 1). In addition, the majority of the studies were serological in nature (17/19; 89.5%), except for two (10.5%) studies that reported genetic detection of WNV from mosquitoes and crocodiles (Table 1). This points to the need for increased genomic surveillance of mosquito-borne pathogens in vectors, animals and humans for a better understanding of the molecular epidemiology of these diseases (Table 2). Considering the limited number of reports documenting clinical cases, establishment of facility-based surveillance of vector-borne zoonoses in health facilities could provide valuable insights on disease severity, mortalities associated with infections and immune responses. As such, the need for building laboratory capacity for routine screening of these pathogens as well as training laboratory personnel cannot be overemphasised. Vector ecology studies aimed at understanding the drivers of vector abundance, pathogen host range (i.e., including the range of vectors and reservoirs), parasite-host

**Table 1. Studies reporting the serologic and/or molecular detection of vector-borne pathogens either in vectors and/or animals including humans in Zambia.**

| Vector-borne Pathogen | Study carried out in: | | |
|---|---|---|---|
| | **Vectors** | **Animals** | **Humans** |
| **Mosquito-borne** | | | |
| Yellow fever virus (YVF) | | | #12,13 |
| Ntaya virus (NTAV) | | | #15 |
| Dengue virus (DENV) | | | #23,24 |
| Zika virus ZIKV) | | #28 | #24,27 |
| West Nile virus (WNV) | †31 | †32 | #30 |
| Chikungunya virus (CHIKV) | | | #24,35 |
| Mayaro virus (MAYV) | | | #24 |
| Sindbis virus (SINV) | | | #36 |
| Rift valley fever virus (RVFV) | | #39,40,42 | |
| **Tick-borne** | | | |
| Crimean–Congo haemorrhagic fever virus (CCHFV) | †47 | #47 | |
| *Anaplasma* spp. | | †49,50,51 | |
| *Ehrlichia* spp. | | †51 | |
| *Rickettsia* spp | †52,53,54,55 | †56 | #57 |
| *Borrelia* spp | †58 | †58 | †58 |
| *Coxiella burnetii* | | †60,61 | #57 |
| **Flea-borne** | | | |
| *Yersinia pestis* | †65,69,70 | ‡70 | †69 |
| *Rickettsia felis* | †67 | †67 | |
| *Rickettsia asembonensis* | †67 | | †79 |
| **Tsetse-borne** | | | |
| Zoonotic *Trypanosoma sp* | | †82,83,84,85 | †86,87 |

Notes: The † symbol in the table denote molecular detection; the # symbol denote serologic detection and the ‡ symbol denotes studies with both molecular and serologic detection. The numbers next to the symbols are reference list numbers of studies that reported the detection of vector-borne pathogens in Zambia.

**Table 2. Research priority areas for vector-borne zoonoses in Zambia requiring research funding.**

| Vector-borne zoonoses | Research priority areas | | |
|---|---|---|---|
| | **Surveillance** | **Diagnostics** | **Capacity-building** |
| Mosquito-borne | • Increased genomic surveillance of pathogens in vectors, reservoir hosts, companion animals and domestic livestock to improve the molecular epidemiology comprehension and development of control strategies.<br>• Increased serological surveillance covering new geographic areas where competent vectors exist.<br>• Increased clinical surveillance of the pathogens in known endemic regions.<br>• Vector ecological studies to inform their distribution | • Utilization of metagenomics next generation sequencing for detection, identification and characterization of known and novel pathogens in clinical specimens, mosquitoes, wild and domestic animals.<br>• Development and deployment of affordable viral antigen capture rapid diagnostic kits for use in clinical settings<br>• Improvement of sensitivity and specificity of serological assays | • Capacity building for improved and routine diagnostic screening in public healthcare facilities.<br>• Capacity building for remote sensing, risk-based mapping and modelling to support vector ecological studies<br>• Capacity building in genomic surveillance and bioinformatics for pathogen detection, identification and characterization |
| Tick-borne | • Serological, genomic and clinical surveillance of viral tick-borne pathogens in humans and other reservoir hosts.<br>• Serological, genomic and clinical surveillance of bacterial tick-borne pathogens in humans and other reservoir hosts.<br>• Genomic surveillance of pathogens in competent vectors<br>• Molecular epidemiological studies to inform control strategies<br>• Vector ecological studies to inform their distribution | • Utilization of metagenomics next generation sequencing for detection, identification and characterization of known and novel pathogens in clinical specimens, vectors, wild and domestic animals.<br>• Development of low-cost multiplex molecular tools for detection of tick-borne pathogens of veterinary and medical importance. | • Capacity building for improved and routine diagnostic screening in public healthcare facilities.<br>• Capacity building for remote sensing, risk-based mapping and modelling to support vector ecological studies<br>• Capacity building in genomic surveillance and bioinformatics for pathogen detection, identification and characterization |
| Flea-borne | • Heightened awareness about risk factors for flea-borne zoonoses among high-risk populations.<br>• Vector ecological studies to inform their distribution | • Development and deployment of affordable real-time PCR assays for detection of multiple genes of Rickettsia in clinical and vector specimens, thus negating the need for sequencing for pathogen identification<br>• Utilization of metagenomics next generation sequencing for detection, identification and characterization of known and novel pathogens in clinical specimens, vectors, wild and domestic animals. | • Capacity building for improved and routine diagnostic screening in public healthcare facilities.<br>• Capacity building for remote sensing, risk-based mapping and modelling to support vector ecological studies<br>• Capacity building in genomic surveillance and bioinformatics for pathogen detection, identification and characterization<br>• Establish specific reference laboratories for detection and identification of flea-borne pathogens. |
| Tsetse-borne | • Serological and genomic surveillance in humans in endemic areas<br>• Continued surveillance in wildlife, livestock, and pets in tsetse-infested regions<br>• Vector ecological studies to inform their distribution<br>• Establishment of a national Human African Trypanosomiasis surveillance and tsetse control programs | • Validation and utilization of molecular techniques (e.g. PCR and LAMP) which are more sensitive than microscopy for passive case detection in primary care centres.<br>• Application of molecular techniques, including next generation sequencing for pathogen detection, identification, and characterization from vectors, livestock, pets and humans. | • Development of laboratory and healthcare personnel capacity for improved and routine diagnostic screening in public healthcare facilities, particularly in tsetse-infested regions.<br>• Capacity building for remote sensing, risk-based mapping and modelling to support vector ecological studies<br>• Capacity building in genomic surveillance and bioinformatics for pathogen detection, identification and characterization |

interactions and factors influencing frequency of human-vector contacts should be prioritized. Remote sensing and risk-based mapping are other research areas that are needed to inform appropriate interventions of vector-borne zoonoses.

## Discussion

This review sought to examine the current knowledge of vector-borne zoonotic pathogens in Zambia. Overall, the study found considerable evidence of presence and active circulation of

these pathogens among vectors, wildlife and domestic animals and humans. Strikingly, reports of human clinical cases attributable to vector-borne pathogens were very limited. This finding could suggest a lack of diagnostic capacity for detection and identification of these infections in many health facilities in the country, a situation that has derailed a better appreciation of the epidemiological picture of vector-borne zoonoses in Zambia, as well as in many African countries [5].

The studies on mosquito-borne viruses revealed that North-Western, Western, Central, Copperbelt and Southern provinces of Zambia could be hotspots for human infections and should be targeted for both field and facility-based surveillance. In some areas, several mosquito-borne viruses were detected [24,27]; this co-circulation has the potential to lead to co-infections, thus presenting a challenge for diagnostic and treatment options in the event of clinical disease as observed elsewhere [96]. With several outbreaks of mosquito-borne viruses being reported in Africa and Southern Africa in particular, along with overlapping ecological niches for the vectors, coupled with the serological and genetic evidence that has accumulated so far, the threat of mosquito-borne arboviral epidemics in Zambia seems possible [10,19,26,35,44,97].

Many studies reviewed in the present study were serological and there was evidence of co-circulation. However, considering that human arboviral infections especially flavivirus infections induce the production of cross-reactive antibodies, often making serology inconclusive, it is important that results from these studies are interpreted cautiously. In fact, this is further complicated by a patient's previous flavivirus exposure, particularly in regions where multiple antigenically related flaviviruses co-circulate and makes it difficult to attribute human disease burden to specific flaviviruses [98,99]. We therefore strongly recommend for increased genomic surveillance of these pathogens to provide precise molecular epidemiological data that will complement the serological data. These data will be necessary to drive enactment of relevant policies that will tackle vector-borne diseases in Zambia. For instance, to be cost effective, the genomic surveillance could utilize banked sera in hospitals or residual blood specimens from routine diagnostic screening of other diseases and could use consensus screening tools such as multiplex PCRs to increase on detection rates. The genomic surveillance could also target companion animals and livestock sharing habitats with humans and essentially using them as sentinels for arboviral circulation in human populations.

With regards to risk factors, exposure of humans to vectors was the main risk factor associated with mosquito-borne arboviral infections reported in Zambia [13]. This included exposure to suitable breeding microhabitats for mosquitoes that are recognized as arboviral vectors. For instance, *Aedes* spp. mosquitoes are known to inhabit flowering pots around homes, water holding containers or unattended containers [100]. This phenomenon in Zambia was shown by Masaninga et al., (2016) who collected vector mosquitoes (*Culex* and *Aedes* spp.) from outdoor containers, in urban regions of North-Western and Western provinces of Zambia where some zoonotic arboviruses have been documented through serological studies [101]. Humans are usually unaware of such microhabitats and this may enhance or sustain the arboviral infections as the vectors are sustained by suitable breeding habitats as well as sufficient blood-meals from humans. In this regard, awareness efforts are fundamental and could focus on educating communities on the impact of such microhabitats. In addition, economic activities such as fishing were strongly associated with high serological positivity for viruses such as CHIKV in humans [24]. Furthermore, high rainfall was linked to RVFV distribution in Zambia as previously observed in Mozambique and South Africa [40,44,45].

Recently, compelling evidence of circulation of CCHFV in cattle and ticks in Zambia was provided [47], supporting the idea that CCHFV is endemic in most Southern African countries and is the most recognised tick-borne viral zoonoses globally [46]. In fact, most

neighbouring countries of Zambia are endemic to CCHFV [46]. With the presence of hard ticks in Zambia and the overlapping ecological habitats of these ticks across countries, the movement of people across the neighbouring countries may further exacerbate the zoonotic risk and establishment of CCHFV in the local ticks. [102]. It is important for future studies to now focus on investigating whether CCHFV circulates in humans, and even more so in areas where the vector ticks are endemic (Table 2).

We noted several reports of tick-borne bacterial zoonoses in livestock, pet dogs and wildlife. Though no human cases have been recorded, the detection of zoonotic *Rickettsia* and *Anaplasma* in NHPs and domestic dogs is concerning. Sequences of the detected *Rickettsia* in NHPs were similar to the *Rickettsia* sp. causing African tick bite fever in sub-Saharan Africa [49,103]. In addition, *Rickettsia* sp. were recently detected in ticks collected from cattle in southern Zambia suggesting the wide spread presence of the bacterium for African tick bite fever in Zambia [55]. Furthermore, the detected *Anaplasma* sp. (*A. phagocytophilum* and *A. platys*) from NHPs and dogs are known causative agents of human anaplasmosis [104]. Taken together, these reports suggest the active circulation of agents of African tick bite fever and human anaplasmosis in the sylvatic and urban cycles with a high potential to infect humans. Human rickettsiosis and anaplasmosis are emerging tick-borne zoonoses globally, causing fever related illnesses that are usually difficult to diagnose as most healthcare providers are unaware of them coupled with limited diagnostic capacity in most healthcare facilities of African countries [48]. Equally, the surveillance of *C. burnetii* should be scaled up in humans. From the reports we reviewed, we observed that the pathogen was detected in livestock, rodents and dogs, increasing the risk for human infection due to relative closeness of these animal hosts to humans. However, it remains unknown whether there is an active zoonotic transmission going on between the infected livestock and humans, as genome detection and clinical disease in humans has never been reported. Though only serological evidence of human infection with *C. burnetii* has been reported in the country, the infection is common with high prevalence rates and has been detected in a wide range of animal species across Africa [105].

Among the flea-borne pathogens reported in Zambia, the epidemiology and ecology of *Y. pestis* in Zambia appears to be fairly well known, though strategies on outbreak prevention and control still remain unclear. On the other hand, *R. felis* epidemiology requires elucidation of possible human infections as well as identification of endemic regions in order to devise effective control measures. Additionally, *R. asembonensis*, a close relative to *R. felis* has been reported in cat fleas collected from dogs as well as in human blood [79]. Further epidemiological studies are still necessary for the understanding of flea-borne zoonoses in Zambia. This is in a bid to put in place effective control measures as well as inform public health policies to curtail the emergence of flea-borne epidemics.

From the few HAT cases that has been reported so far, it is evidently clear that HAT is a neglected zoonosis in Zambia. Most of the earlier studies focussed more on the distribution of the vector and associated ecological factors. On the disease part, many studies done in Zambia and elsewhere [81] focussed more on African Animal Trypanosomiasis due to its associated economic losses neglecting the zoonotic potential of trypanosomes adapted to cause human disease. Only three reports were found to be describing HAT in humans based on passive surveillance [86,87,89], pointing to the lack of deliberate policies to improve diagnosis of HAT in the Zambian healthcare system. Concerted efforts are therefore required from all key players to ensure that active surveillance of human infective trypanosomes is mounted in order to improve the detection of HAT in Zambia. In fact, the detections of zoonotic trypanosomes in pet dogs compounds the problem of this neglected tropical disease as indicated by the ease of transmission from the sylvatic cycle to the urban cycle and requires timely interventions [83–85].

We also noted with concern the lack of data on vector control policies aimed at combating vector-borne zoonoses in Zambia. The mosquito control programme that has been ongoing for the past two decades is mainly indoor residual spraying and long-lasting insecticidal nets (LLIN) approaches targeting mosquito species transmitting malaria [106]. Though beneficial indirectly, to a large extent, it excludes other outdoor mosquito vectors that transmit vector-borne zoonoses. Similarly, tick and tsetse control in Zambia has been implemented in light of controlling livestock diseases [107,108]. In addition, flea control is largely non-existent in Zambia except perhaps through spraying of pets for ectoparasite control at individual household level. Indeed, the design of such vector control strategies has been one without a One Health framework in mind leaving vector-borne zoonoses unattended to from the vector control perspective. From the above, it is evident that there is an urgent need to develop national vector control policies and frameworks that will specifically target the control of vector-borne zoonoses in Zambia.

In addition, investment into studies such as remote-sensing based risk mapping has also the potential to improve the understanding of all the above-mentioned vector-borne zoonoses in Zambia. Such technology will help estimate how environmental variables such as humidity, precipitation, temperature and ground cover influence vector abundance, infection rates and interspecies transmission [109]. For example, changes in climate have been associated with increased transmission of vector-borne diseases [2]. Furthermore, it is widely recognized that vectors are sensitive to changes in temperature and precipitation [110]. Similarly, as discussed by Mills et al., (2010), variations in climate play a role in the emergence of vector-borne zoonoses because as the population density of vectors increases, the frequency of contact between vectors, humans or other hosts will also increase [111]. Therefore, application of remote-sensing based risk mapping will be critical in understanding the influence of climate change in the emergence of vector-borne zoonoses in Zambia (Table 2).

## Conclusions

Overall, this review has shown that diagnosis of vector-borne zoonotic infections in Zambian healthcare facilities is very limited, if any. Most reports in humans were serological in nature and potentially limited in their interpretation to infer actual disease burden in humans owing to the cross-reactivity of arboviruses. In contrast, molecular as well as serological studies have shown the circulation of these pathogens in vectors and/or wildlife and domestic animals and suggest that infections and possibly clinical disease in humans could be going on undetected due to limited laboratory diagnostic capacity. Speculatively, it is our considered opinion that there is low awareness of emerging and re-emerging zoonoses in Zambia among healthcare practitioners. With glaring evidence provided in this review on the active circulation and transmission of vector-borne zoonotic pathogens across various invertebrate and vertebrate hosts, the need for establishment of policy frameworks towards tackling neglected vector-borne zoonoses cannot be overemphasized. It is our considered opinion that deliberate policies for laboratory strengthening to support routine screening of these pathogens in health facilities are urgently needed for improved diagnosis and management of zoonoses in the country.

*De facto*, scaling up of research into emerging and re-emerging zoonotic pathogens will form the basis for informing these interventions and addressing the existing knowledge gaps. Hence, there is an urgent need for the research community in Zambia and its cooperating partners to urgently scale up One Health research on vector-borne and emerging and re-emerging zoonoses not only to inform policies that could translate into improved healthcare provision, but also enable the country to prepare for future epidemics, including pandemics.

## Author Contributions

**Conceptualization:** Monicah M. Mburu, Katendi Changula, Walter Muleya, Catherine G. Sutcliffe, Douglas E. Norris, Philip E. Thuma, Phillimon Ndubani, Simbarashe Chitanga, Edgar Simulundu.

**Data curation:** Benjamin Mubemba, Monicah M. Mburu, Lavel C. Moonga, Edgar Simulundu.

**Formal analysis:** Benjamin Mubemba, Monicah M. Mburu, Lavel C. Moonga, Edgar Simulundu.

**Investigation:** Benjamin Mubemba, Edgar Simulundu.

**Methodology:** Benjamin Mubemba, Edgar Simulundu.

**Project administration:** Masahiro Kajihara, Hirofumi Sawa, Ayato Takada, Edgar Simulundu.

**Resources:** Hirofumi Sawa, Ayato Takada.

**Supervision:** Edgar Simulundu.

**Visualization:** Benjamin Mubemba, Edgar Simulundu.

**Writing – original draft:** Benjamin Mubemba, Monicah M. Mburu, Lavel C. Moonga, Edgar Simulundu.

**Writing – review & editing:** Benjamin Mubemba, Monicah M. Mburu, Katendi Changula, Walter Muleya, Lavel C. Moonga, Herman M. Chambaro, Masahiro Kajihara, Yongjin Qiu, Yasuko Orba, Kyoko Hayashida, Catherine G. Sutcliffe, Douglas E. Norris, Philip E. Thuma, Phillimon Ndubani, Simbarashe Chitanga, Hirofumi Sawa, Ayato Takada, Edgar Simulundu.

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
