## [Decision Letter · Decision Letter 0]

12 Oct 2021

Dear Dr Simulundu,

Thank you very much for submitting your manuscript "Current knowledge of vector-borne zoonotic pathogens in Zambia: A clarion call to scaling-up “One Health” research in the wake of emerging and re-emerging infectious diseases" for consideration at PLOS Neglected Tropical Diseases. As with all papers reviewed by the journal, your manuscript was reviewed by members of the editorial board and by several independent reviewers. The reviewers appreciated the attention to an important topic. Based on the reviews, we are likely to accept this manuscript for publication, providing that you modify the manuscript according to the review recommendations. 

Dear Authors,

Thank you for submitting of your manuscript to PLOSNTD. I regret this long evaluation time – multiple experts had been invited before three finally agreed and now sent their reviews (please see below). 

Your methodology was cleary articulated however one of our reviewers suggested to add more criteria. Regardings to your results, Table 1 and 2 should be reformatted in order to clearly show the lack of information available on pathogen detection. Please rewrite your results and separate the discussion part. In addition, the authors should provide more information such as cross-reactivity of antibodies between flaviviruses in serological surveys, discussion on vector control efforts and policy to combat zoonoses in Zambia. The opposite results of two studies should be add more discussion. Finally, the limitation of the study should be clearly described.

Sincerely,

Sirikachorn Tangkawattana, Ph.D.

Associate Editor

Paulo Pimenta

Deputy Editor

Reviewer's Responses to Questions

**Key Review Criteria Required for Acceptance?**

**Methods**

-Are the objectives of the study clearly articulated with a clear testable hypothesis stated?

-Is the study design appropriate to address the stated objectives?

-Is the population clearly described and appropriate for the hypothesis being tested?

-Is the sample size sufficient to ensure adequate power to address the hypothesis being tested?

-Were correct statistical analysis used to support conclusions?

-Are there concerns about ethical or regulatory requirements being met?

Reviewer #1: I have no methodological concerns. The authors execute an appropriately designed scoping review to achieve the stated objective of identifying research gaps for vector-borne zoonotic pathogens in Zambia.

Reviewer #2: As this was a scoping review article, many of the methodology criteria do not apply. However, in areas where the methods are outlined (lines 150-164), it would be nice to see how many articles were identified, and then which exclusionary filters were imposed. Including the number of studies that were removed at each step would also be ideal. For instance, "articles in languages other than English, n = __". Doing so would also provide more evidence that VBD are understudied in Zambia.

Reviewer #3: -The objectives of the study was clearly articulated a with the study design appropriately addressing the stated objectives.

Adequate literature was reviewed to make up sufficient sample size to ensure adequate power to address the hypothesis being tested.

**Results**

-Does the analysis presented match the analysis plan?

-Are the results clearly and completely presented?

-Are the figures (Tables, Images) of sufficient quality for clarity?

Reviewer #1: Lines 250-254: “Recently, a study conducted in 2016 among residents of Lukanga swamps in Central Province, revealed a 36.9% (79/214) sero-positivity for CHIKV and 19.6% (42/214) for Mayaro virus (MAYV) [24]. A recent study that screened for alphaviruses using 9,699 mosquitoes collected in Zambia (2014-2017) could not detect known alphaviruses (CHIKV, O'nyong'nyong virus, SINV etc.) [37].” Taken together, these results are quite striking – the two studies report essentially opposite findings. The authors should provide additional commentary on the research gap that is highlighted by these contrasting results, and propose hypotheses as to why such high seropositivity is observed near Lukanga, while the collection of nearly 10,000 mosquitos in Zambia resulted in no alphavirus detection.

Reviewer #2: Would recommend reformatting of Tables 1 & 2, which I outline in more detail within my general comments. Ultimately, generating some form of table or figure clearly showing how many studies were detected for each category (serologic vs. pathogen detection) and either stratifying on the basis of vector (as they have done) or pathogen type (virus, bacteria, parasite, etc) would be helpful. This would also highlight the lack of information available on pathogen detection as compared to serologic detection.

Reviewer #3: The analysis presented match the analysis plan however a substantial part of the " Results" as presented was suppose to have been captured under "Discussion".

From line 422 to the end should have been recommendation not results.

**Conclusions**

-Are the conclusions supported by the data presented?

-Are the limitations of analysis clearly described?

-Do the authors discuss how these data can be helpful to advance our understanding of the topic under study?

-Is public health relevance addressed?

Reviewer #1: Lines 427-429: The authors correctly highlight a weakness of the existing literature for Zambia, that “the majority of the studies were serological in nature.” Particularly because of their focus on arboviruses, the authors should highlight the extensive and well-documented cross-reactivity of antibodies between flaviviruses of different species. This represents a major caveat for interpreting any epidemiological studies that rely solely on flaviviral serosurvey data. The poor specificity of flaviviral serologic tests makes it quite challenging to derive accurate estimates of disease attributable to specific flaviviral species. Highlighting this methodological weakness of serosurveys further underscores the authors’ call “for increased genomic surveillance” (Line 430) as a complementary experimental approach. 

In the present manuscript, there is no discussion of vector control efforts to combat zoonoses in Zambia, however understanding Zambia’s history of vector control is necessary to contextualize the reviewed studies. To this end, the authors should include a brief discussion of major vector control policies and efforts/campaigns (insecticide spraying, etc.) that have taken place in Zambia and focused on zoonotic pathogens, particularly if there are studies that measured vector populations or disease incidence pre- and post- control interventions. (Even if there are no published studies, such data may be available from the Ministry of Health.) If there are no such studies, the authors should explicitly highlight this as a research gap.

Reviewer #2: Yes, though as I outline in other parts of my review, I would like to see Tables 1 & 2 modified to highlight at higher resolution where the knowledge gaps lie, and perhaps use that more granular analysis to present some detailed/specific multi-pronged VBD surveillance strategies. Including some preliminary policy recommendations, or at least identifying funding priorities, would be helpful to begin actualizing or making tangible some of the actions that could be taken to enhance our understanding of VBD dynamics in Zambia.

Reviewer #3: The study did not caption "conclusions" notwithstanding some of the findings of the study were supported by the data presented while some were not. Example of the not part of the study: The conclusion that the sparse capacity for diagnosis of vector borne pathogens is as a result of Zambian healthcare facilities being very limited, if any exist at all.

The limitations of analysis were not clearly described

The authors discuss how these data can be helpful to advance our understanding of the topic under study with public health relevance addressed

**Editorial and Data Presentation Modifications?**

Reviewer #1: None.

Reviewer #2: Recommend reformatting Tables 1 & 2; please see Summary & General Comments. Are all papers that were reviewed cited somewhere in the paper? Rather, is it possible to find all papers covered within this scoping review within the references list?

Reviewer #3: (No Response)

**Summary and General Comments**

Reviewer #1: This manuscript by Mubemba et al. is a scoping review of vector-borne zoonotic pathogens in Zambia, with a focus on pathogens transmitted by mosquitos (particularly arboviruses), ticks, fleas, and tsetse flies. Overall, this is a strong, focused manuscript that achieves its stated objective of highlighting research gaps for vector-borne zoonoses in Zambia. The review also highlights some interesting emerging research areas specific to Zambia, particularly Q fever, Yersinia pestis, and multiple pathogens that were discovered in Zambia. As such, this manuscript provides an important foundation for setting research and public health priorities for these pathogens in Zambia and in sub-Saharan Africa more broadly. There are some minor typographical and grammatical errors that will need to be corrected before final publication, however overall the manuscript covers a diverse array of studies while summarizing them clearly and succinctly. My critiques are minor and focus on areas of the Discussion that should be expanded upon by the authors to emphasize research gaps, in support their main thesis. 

I recommend Minor Revisions before the manuscript is accepted for publication.

Reviewer #2: This scoping review examined current knowledge and identified research gaps of vector-borne zoonotic pathogens in Zambia. The authors used several major scientific databases and a selected set of search terms to identify the presence of articles on this topic. Major findings included that there was a lack of representation of articles describing infection by means of pathogen isolation or detection of nucleic acid. Conversely, most articles to date have provided information on serological evidence of infection. Serological evidence is useful for identifying past exposure, but these data are limited in their interpretation. The authors have made good note of this, as they state that human burden of disease is difficult to attribute to any of these pathogens when the only information we have on host-pathogen interaction is seroconversion/evidence of exposure. Knowledge gaps in this area make implementation of national policy and advocating for transmission control funding difficult. 

The authors have compiled a compelling body of work, highlighting the major knowledge gaps surrounding vector-borne disease circulation in Zambia. I really enjoyed reading this piece and agree with the authors on many points. A few areas worth building out/commenting on:

Related to flaviviruses, there is a high degree of serologic cross-reactivity, so by pinpointing that the majority of what we know about mosquito-borne flavivirus circulation in Zambia is based on serologic evidence is concerning for many reasons - one being that in general, flavivirus serology is not very reliable (for many cases, and this is dependent upon the diagnostic platform being used).

I would recommend re-formatting Table 1 to highlight which of the studies indicate serologic evidence of infection and which describe genomic detection. You might consider having a darkened circle for genomic detection, open circle for serologic, and half-filled circle for both. You could even list the reference numbers in the box alongside the circle, which would allow for easier navigation to those articles of interest by the reader, and also eliminate the fifth column in the table. 

I appreciate the suggestions for increasing genomic surveillance. This portion of the discussion could be built out a bit more with specific suggestions for how to do this in a cost-effective manner. For instance, using banked samples (for instance, does Zambia National Blood Transfusion Service (ZNBTS) have banked sera?) and consensus screening tools (pan-flavivirus, for instance) to maximize detection rates. Further, you might recommend increasing genomic surveillance not only in vectors and reservoir hosts, but in companion animals and domestic livestock that share habitat and interact frequently with humans - essentially, using them as sentinels for arboviral circulation in human populations. While there are limitations to this approach with regard to arthropod bloodmeal/host preference, this would allow for an analysis into which pathogens are being transmitted by arthropods in the same geographic region. 

I would also build out Table 2 in a way that stratifies the suggestions. You could stratify each row by capacity-building, diagnostics, and surveillance… and then make recommendations for priority funding areas if that is possible. You might also consider stratifying each row on the basis of geographic regions of coverage… so for instance, “hospital/focal surveillance”, “regional surveillance”, or more integrated national surveillance programs to track pathogen spread and evolution. Another stratification could be “immediate”, “short-term”, and “long-term”, where more immediate recommendations might include some of the education/awareness campaigns you outline in the discussion. These are just a few ideas, and I suspect you could do this in a number of different ways to help drive home the impact of these recommendations. Table 2 is really the big take-home from this paper and could be featured in a way that allows for strong identification of patterns and then building off of that, some solid policy recommendations, or at least identification of funding priorities.

Reviewer #3: -Why was the focus only on mosquitoes, ticks, fleas and tsetsefly as vectors and not others such as blackfly etc

- Line 126 to 129 should be referenced otherwise it is a serious assertion to make if unsubstantiated

-On line 191, it is stated that Culex mosquito is a "suspected " but not a confirmed vector of NTAV. Which is the confirmed one then

-Which other countries have been reported for NTAV, dengue, zika, WNV, Chikungunya, RVF, tick borne bacterial diseases and flea borne zoonotic pathogens as reported for yellow fever which is in 32 other tropical countries aside Zambia

-When was the first case of encephalitis, zika and CCHF recorded in Zambia

-Are there no mosquito borne bacteria infections' study in Zambia?

-Which of the infection in line 274 and 275 is line 276-277 describing its symptoms?

-What are the symptoms of borreliosis and Q-fever

-Edit the name of the Table 1 by removing the first words which is 'A summary table showing the number of' and add of Zambia after the 'humans'

- A footer should be shown below Table 1 to show what ' X ' signifies

-Give reference to the sentence on line 465

- On line 468, it is stated that '...should be targeted for both field and facility-based surveillance efforts' – remove the word 'effort'

- Give reference to the sentences in lines 474 to 476

- On line 576-578, the sentence is speculative and should be stated as such

- Acknowledgments on line 586 is spelt wrongly

-The 'results' section had a lot of discussion which should be moved to the 'discussion' section. Rewrite your results and discussion.

PLOS authors have the option to publish the peer review history of their article (what does this mean?). If published, this will include your full peer review and any attached files.

Reviewer #1: Yes: Joshua R. Lacsina

Reviewer #2: Yes: Anna Fagre

Reviewer #3: No

Figure Files:

Data Requirements:

Reproducibility:

References

---

## [Decision Letter · Decision Letter 1]

18 Jan 2022

Dear Dr Simulundu,

Thank you very much for submitting your manuscript "Current knowledge of vector-borne zoonotic pathogens in Zambia: A clarion call to scaling-up “One Health” research in the wake of emerging and re-emerging infectious diseases" for consideration at PLOS Neglected Tropical Diseases. As with all papers reviewed by the journal, your manuscript was reviewed by members of the editorial board and by several independent reviewers. The reviewers appreciated the attention to an important topic. Based on the reviews, we are likely to accept this manuscript for publication, providing that you modify the manuscript according to the review recommendations. 

Sincerely,

Sirikachorn Tangkawattana, Ph.D.

Associate Editor

Paulo Pimenta

Deputy Editor

Reviewer's Responses to Questions

**Key Review Criteria Required for Acceptance?**

**Methods**

-Are the objectives of the study clearly articulated with a clear testable hypothesis stated?

-Is the study design appropriate to address the stated objectives?

-Is the population clearly described and appropriate for the hypothesis being tested?

-Is the sample size sufficient to ensure adequate power to address the hypothesis being tested?

-Were correct statistical analysis used to support conclusions?

-Are there concerns about ethical or regulatory requirements being met?

Reviewer #1: No concerns.

Reviewer #2: Table 1 is much improved with the stratification of publications based on whether detection was serologic or molecular in nature.

Reviewer #3: (No Response)

**Results**

-Does the analysis presented match the analysis plan?

-Are the results clearly and completely presented?

-Are the figures (Tables, Images) of sufficient quality for clarity?

Reviewer #1: No concerns.

Reviewer #2: It is striking to me how unevenly distributed Table 1 is. Look at the number of papers providing evidence of exposure to mosquito-borne arboviruses in humans compared to parallel data available on animals and vectors. It appears that surveillance of mosquito-borne viruses in vectors and animals is hugely lacking -- and those data are critical when considering these transmission cycles through a One Health lens. Conversely, there are MANY papers demonstrating molecular detection of bacterial/protozoal tick-borne, flea-borne, and tsetse-borne diseases in animals and arthropod vectors, and to a lesser extent in humans.

Reviewer #3: (No Response)

**Conclusions**

-Are the conclusions supported by the data presented?

-Are the limitations of analysis clearly described?

-Do the authors discuss how these data can be helpful to advance our understanding of the topic under study?

-Is public health relevance addressed?

Reviewer #1: No concerns.

Reviewer #2: I think discussing how targeting only one aspect of the One Health triad (or in this case, one aspect of the transmission cycle) risks over-interpretation of the role that component plays in the cycle, and the general imbalance between studies suggests that this pattern may translate to our overall understanding of these processes on a larger scale. I see now that you’ve included in the beginning of the conclusions that it’s problematic that a majority of human-focused studies contained only serologic evidence, complicating the interpretation of this data (or ability to associate it with disease burden, levels of shedding/viremia, etc.) However, I think it’s worth elaborating on the point you make in lines 146-149. WHY is there such an imbalance in molecular detections? Sampling bias?

Reviewer #3: (No Response)

**Editorial and Data Presentation Modifications?**

Reviewer #1: No concerns.

Reviewer #2: Line items: 

• On page 19 of document, lines 419-422, it appears something happened with the ‘track changes’, as I believe the last sentence of that section is fragmented.

• Missing reference in page 2 of the discussion, and page 6 (line 118, 121) of discussion,

• Line 123: ‘one-health’ is used but in the rest of the paper, you use One-Health (capitalized). Also, I think I’ve seen it used more often in the past without a hyphen, but I’ll defer to the editor on that.

• There were some comments in the conclusions (lines 144-145) that I found a bit confusing.

Reviewer #3: (No Response)

**Summary and General Comments**

Reviewer #1: This manuscript by Mubemba et al. is a scoping review of vector-borne zoonotic pathogens in Zambia, with a focus on pathogens transmitted by mosquitos (particularly arboviruses), ticks, fleas, and tsetse flies. This work provides an important foundation for setting research and public health priorities for these pathogens in Zambia and in sub-Saharan Africa more broadly. While there remain some minor typographical and grammatical errors that will require editing before final publication, in terms of content, the authors have fully addressed all my critiques. Indeed, the authors' revisions to Table 1 (based on the suggestions of Reviewer #2) have markedly improved the clarity and visual presentation of the literature review. I therefore support acceptance of the manuscript for publication.

Reviewer #2: (No Response)

Reviewer #3: All scientific names should be italized

PLOS authors have the option to publish the peer review history of their article (what does this mean?). If published, this will include your full peer review and any attached files.

Reviewer #1: Yes: Joshua R. Lacsina

Reviewer #2: Yes: Anna C Fagre

Reviewer #3: No

Figure Files:

Data Requirements:

Reproducibility:

References

---

## [Editor Report · Decision Letter 2]

24 Jan 2022

Dear Dr Simulundu,

We are pleased to inform you that your manuscript 'Current knowledge of vector-borne zoonotic pathogens in Zambia: A clarion call to scaling-up “One Health” research in the wake of emerging and re-emerging infectious diseases' has been provisionally accepted for publication in PLOS Neglected Tropical Diseases.

Best regards,

Sirikachorn Tangkawattana, Ph.D.

Associate Editor

Paulo Pimenta

Deputy Editor

---

## [Editor Report · Acceptance letter]

28 Jan 2022

Dear Dr Simulundu,

We are delighted to inform you that your manuscript, "Current knowledge of vector-borne zoonotic pathogens in Zambia: A clarion call to scaling-up “One Health” research in the wake of emerging and re-emerging infectious diseases," has been formally accepted for publication in PLOS Neglected Tropical Diseases.

Best regards,

Shaden Kamhawi

co-Editor-in-Chief

Paul Brindley

co-Editor-in-Chief
